**Data Availability Statement:** All relevant data are within the paper.

**Funding:** Institutional support and physical resources were provided by the University Witten/

# Observational study of changes in utilization and outcomes in mechanical ventilation in COVID-19

Christian Karagiannidis[1]☉*, Corinna Hentschker[2]☉, Michael Westhoff[3,4], Steffen Weber-Carstens[5], Uwe Janssens[6], Stefan Kluge[7], Michael Pfeifer[8,9], Claudia Spies[5], Tobias Welte[10], Rolf Rossaint[11], Carina Mostert[2]☉, Wolfram Windisch[1]☉

**1** Department of Pneumology and Critical Care Medicine, Cologne-Merheim Hospital, ARDS and ECMO Center, Kliniken der Stadt Köln, Witten/Herdecke University Hospital, Cologne, Germany, **2** Research Institute of the Local Health Care Funds, Berlin, Germany, **3** Department of Pneumology, Sleep and Critical Care Medicine, Lungenklinik Hemer, Hemer, Germany, **4** University Witten/Herdecke, Witten, Germany, **5** Department of Anesthesiology and Operative Intensive Care Medicine (CCM, CVK), Charité - Universitätsmedizin Berlin, Berlin, Germany, **6** Medical Clinic and Medical Intensive Care Medicine, St.-Antonius Hospital, Eschweiler, Germany, **7** University Medical Center Hamburg-Eppendorf, Hamburg, Germany, **8** Department of Internal Medicine II, University Hospital Regensburg, Regensburg, Germany, **9** Department of Pneumology, Donaustauf Hospital, Donaustauf, Germany, **10** Department of Respiratory Medicine and German Centre of Lung Research (DZL), Hannover Medical School, Hannover, Germany, **11** Department of Anesthesiology, University Hospital Aachen, RWTH Aachen, Aachen, Germany

☉ These authors contributed equally to this work.
* Christian.Karagiannidis@uni-wh.de, karagiannidisc@kliniken-koeln.de

## Abstract

### Background

The role of non-invasive ventilation (NIV) in severe COVID-19 remains a matter of debate. Therefore, the utilization and outcome of NIV in COVID-19 in an unbiased cohort was determined.

### Aim

The aim was to provide a detailed account of hospitalized COVID-19 patients requiring non-invasive ventilation during their hospital stay. Furthermore, differences of patients treated with NIV between the first and second wave are explored.

### Methods

Confirmed COVID-19 cases of claims data of the Local Health Care Funds with non-invasive and/or invasive mechanical ventilation (MV) in the spring and autumn pandemic period in 2020 were comparable analysed.

### Results

Nationwide cohort of 17.023 cases (median/IQR age 71/61–80 years, 64% male) 7235 (42.5%) patients primarily received IMV without NIV, 4469 (26.3%) patients received NIV without subsequent intubation, and 3472 (20.4%) patients had NIV failure (NIV-F), defined

Herdecke and Kliniken der Stadt Köln and the Federal Association of the Local Health Care Funds. The funders had no role in study design, data collection and analysis, decision to publish, or preparation of the manuscript.

**Competing interests:** Dr. Karagiannidis reports personal fees from Maquet, personal fees from Xenios, personal fees from Bayer, non-financial support from Speaker of the German register of ICUs, grants from German Ministry of Research and Education, during the conduct of the study. Dr. Hentschker has nothing to disclose. Dr. Westhoff has nothing to disclose. Dr. Weber-Carstens has nothing to disclose. Dr. Janssens has nothing to disclose. Dr. Kluge reports non-financial support from Ambu, ETView Ltd, Fisher & Paykel and Xenios., grants from Daiichi Sankyo, Pfizer, personal fees from Astra, C.R. Bard, Baxter, Biotest, Cytosorbents, Fresenius, Gilead, MSD, Pfizer, Philips, ZOLL, personal fees and other from Bayer, Fresenius, Gilead, MSD und Pfizer, outside the submitted work. MP reports no conflicts of interests in regard to the manuscript, lecture fees from Boehringer, Novartis, Astra_Zeneca, Roche and fees for advisory board meetings from Boehringer, Novartis, Roche Current president of the German Society of Pneumology. Dr. Spies reports grants from Public Grants, grants from IIT grants from companies, other from Meeting support from companies (e.g. for the Leopoldina 2020 meeting), outside the submitted work; In addition, Dr. Spies has a patent EEG monitoring licensed, and a patent Ceilings licensed. Dr. Welte reports grants from German Minstry of Research and Education, during the conduct of the study. This does not alter our adherence to PLOS ONE policies on sharing data and materials.

by subsequent endotracheal intubation. The proportion of patients who received invasive MV decreased from 75% to 37% during the second period. Accordingly, the proportion of patients with NIV exclusively increased from 9% to 30%, and those failing NIV increased from 9% to 23%. Median length of hospital stay decreased from 26 to 21 days, and duration of MV decreased from 11.9 to 7.3 days. The NIV failure rate decreased from 49% to 43%. Overall mortality increased from 51% versus 54%. Mortality was 44% with NIV-only, 54% with IMV and 66% with NIV-F with mortality rates steadily increasing from 62% in early NIV-F (day 1) to 72% in late NIV-F (>4 days).

## Conclusions

Utilization of NIV rapidly increased during the autumn period, which was associated with a reduced duration of MV, but not with overall mortality. High NIV-F rates are associated with increased mortality, particularly in late NIV-F.

## Introduction

Within one year, the SARS-CoV-2 pandemic has affected more than 235 million people worldwide (https://coronavirus.jhu.edu/map.html). Mortality rates of patients requiring ICU treatment are ranging up to over 50% [1–4], depending on the severity of respiratory failure and response to treatment, but also influenced by age, comorbidities and a ceiling of therapeutic interventions [1, 2, 5–7].

Mechanical ventilation (MV) is a life-saving option in severe COVID-19 cases, but mortality rates in patients on MV remain high [4, 5, 8]. Non-invasive ventilation (NIV) is suggested to reduce the complications of invasive MV. The use of noninvasive respiratory support in acute respiratory failure due to viral infection is still debated [9, 10]. For COVID-19 patients, current guidelines recommend stepping up to NIV when oxygenation worsens during oxygen therapy, and to consider intubation if $PaO_2/FiO_2$ is decreased below 150 mmHg [11–13] or the clinical presentation of the patients has worsened [11, 13–17], despite international guidelines being still inhomogeneous in recommendations [18]. In Germany, detailed epidemiological data about what types of interfaces are used are not available, but face masks are the most commonly used interface in acute respiratory failure in clinical practice in Europe [19], while helmets are only rarely used by very few experienced centers. However, global current practices of MV widely differ, also depending on COVID-19-associated limited resources [4, 20, 21]. Therefore, the role of NIV remains a matter of uncertainty and discussion, especially with regard to the balance between the NIV benefits and the risk of NIV failure (NIV-F). The mortality of patients receiving NIV was in a wide range up to 45% [1, 22]. In contrast to this, the mortality rate in patients with NIV-F ranged between 35% and 74% [22–24]. Hence, interpretation of data and obtaining conclusive strategies concerning the optimal and individual timing of intubation remain uncertain [25].

Therefore, the aim of the current study was to determine detailed characteristics and outcomes of 7,490 hospitalized COVID-19 patients with MV on the ICU in a large, unselected and unbiased cohort of patients with confirmed COVID-19 in one of the least resource limited health care systems [26], particularly focusing on patients requiring invasive MV or NIV with specific emphasis on NIV-F. Furthermore, we explored the changes between the first spring and second autumn/winter period.

## Data and methods

The inpatient data of the general local health insurance funds, which cover around a third of the German population, were analyzed. In general, this is a retrospective analysis of claims data from this registry. Data extraction was done by the scientific institute of the health care insurance, whereas analysis was done by the author group. It is an administrative data set containing patient information like age, gender, diagnosis and procedure codes. However, detailed medical information such as laboratory data is not recorded. All cases were included for which admission and discharge dates as well as diagnoses and procedures were coded. Only patients with laboratory-confirmed SARS-CoV-2 infection (diagnosis code U07.1!) were included. The patients were at least 18 years old and were admitted to hospital between February 1, 2020 and November 30, 2020.

The original data structure is at the case level, i.e. insured persons who were transferred to another hospital during their hospital stay appear several times in the data set. Therefore, cases who were transferred during their hospital stay (discharge date of one hospital corresponds to the admission date of another hospital) were merged. Thus, the current analysis was performed at the patient level. The following OPS codes from the German DRG systematic coding were analyzed: 8–701, 8–704, 5–311, 5–313 and 8–706. For continuous variables, we report means with SDs and medians with IQRs. For categorical variables, we report absolute numbers and percentages.

The primary analysis includes all patients with mechanical ventilation, either non-invasive or invasive on the ICU without any missing, but secondary analysis focuses on patients with MV for more than 6 hours, i.e. invasive MV or NIV. These patients were divided into three subgroups: 1) patients with primary invasive MV without any NIV attempt preceding intubation, 2) patients with NIV exclusively, who have not been escalated to intubation, and 3) those with NIV-F, defined by endotracheal intubation following NIV. In the last group, a procedure code for both NIV and invasive MV was assigned. If invasive MV was started on the day following NIV, the patient was assigned to the NIV-F group. If invasive ventilation was started on the same day as NIV initiation, the patient was assigned to the invasive MV. This definition might help to distinguish real NIV establishment from NIV as short preoxygenation before intubation. Both, patients with less than 6 documented hours of ventilation (n = 695; Table 1) and patients with more than 6 documented hours of ventilation but without a corresponding procedure code for NIV or invasive ventilation (n = 305; Table 1) were not assigned to the three subgroups. With the inclusion of the procedural data, it was possible to roughly determine the NIV duration when switching from NIV to invasive MV.

The study was approved by the local ethical committee (University Witten/Herdecke, 92/2020).

## Findings

Between February 1 2020 and February 28 2021, 16.328 hospitalized Covid-19 patients received MV (Table 1). Data of patients treated during the summer months (June to September) is not shown separately in the table due to the relatively low number. Age distribution, sex and the frequency of comorbidities show only slight differences when comparing the two periods of the pandemic as shown in Table 1. The overall median length of hospital stay has decreased from 26 days during the first wave of the pandemic to 21 days during the second wave. This also applies to the overall duration of MV, which decreased from 11.9 to 7.3 days, respectively.

A major difference between the spring and autumn period of the pandemic refers to the application of the different MV modalities. During the first period, 755% of the patients

**Table 1. Patient characteristics comparing the spring and autumn period, ECMO = extracorporeal membrane oxygenation.** The Elixhauser Comorbidity Index is a method of categorizing comorbidities of patients based on the (ICD) diagnosis codes in administrative data.

| Variable | Patients by month of hospital admission | | |
| --- | --- | --- | --- |
| | Total | Admission between February 2020 and May 2020 | Admission between October 2020 and February 2021 |
| **Number of patients** | 17023 | 2376 | 13998 |
| **Age (years)** | | | |
| Mean (SD) | 69.3 (13.1) | 68.1 (13.3) | 69.7 (12.9) |
| Median (IQR) | 71.0 (61.0, 80.0) | 70.0 (60.0, 79.0) | 71.0 (62.0, 80.0) |
| 18–49 years | 1,304 (7.7%) | 204 (8.6%) | 1,010 (7.2%) |
| 50–59 years | 2,328 (13.7%) | 380 (16.0%) | 1,840 (13.1%) |
| 60–69 years | 4,152 (24.4%) | 556 (23.4%) | 3,418 (24.4%) |
| 70–79 year | 4,847 (28.5%) | 726 (30.6%) | 3,956 (28.3%) |
| ≥ 80 years | 4,392 (25.8%) | 510 (21.5%) | 3,774 (27.0%) |
| **Male** | 10,926 (64.2%) | 1,565 (65.9%) | 8,913 (63.7%) |
| **Female** | 6,097 (35.8%) | 811 (34.1%) | 5,085 (36.3%) |
| **Elixhauser comorbidities** | | | |
| Hypertension | 11,708 (68.8%) | 1,545 (65.0%) | 9,717 (69.4%) |
| Diabetes | 7,313 (43.0%) | 947 (39.9%) | 6,097 (43.6%) |
| Cardiac arrhythmias | 7,311 (42.9%) | 1,079 (45.4%) | 5,966 (42.6%) |
| Renal failure | 4,979 (29.2%) | 627 (26.4%) | 4,203 (30.0%) |
| Congestive heart failure | 5,871 (34.5%) | 786 (33.1%) | 4,853 (34.7%) |
| Chronic pulmonary disease | 3,378 (19.8%) | 469 (19.7%) | 2,780 (19.9%) |
| Obesity | 2,659 (15.6%) | 354 (14.9%) | 2,201 (15.7%) |
| Charlson comorbidity index: 0 | 3,118 (18.3%) | 455 (19.1%) | 2,537 (18.1%) |
| Charlson comorbidity index: 1 | 3,495 (20.5%) | 507 (21.3%) | 2,858 (20.4%) |
| Charlson comorbidity index: 2 | 2,814 (16.5%) | 425 (17.9%) | 2,285 (16.3%) |
| Charlson comorbidity index: 3–4 | 4,256 (25.0%) | 558 (23.5%) | 3,564 (25.5%) |
| Charlson comorbidity index: ≥ 5 | 3,340 (19.6%) | 431 (18.1%) | 2,754 (19.7%) |
| **Patients transferred between hospitals** | 5,543 (32.6%) | 856 (36.0%) | 4,373 (31.2%) |
| **Length of hospital stay (days)** | | | |
| Mean (SD) | 32.9 (33.8) | 37.2 (38.7) | 31.1 (31.0) |
| Median (IQR) | 22.0 (13.0, 41.0) | 26.0 (13.0, 49.0) | 21.0 (12.0, 38.0) |
| **Ventilation (days)** | | | |
| Mean (SD) | 13.9 (17.7) | 17.4 (19.2) | 13.1 (17.2) |
| Median (IQR) | 8.0 (2.3, 18.2) | 11.9 (4.8, 23.4) | 7.3 (2.1, 17.2) |
| **Tracheostomy** | 4,017 (23.6%) | 747 (31.4%) | 3,061 (21.9%) |
| **ECMO** | 1,129 (6.6%) | 190 (8.0%) | 864 (6.2%) |
| **Dialysis** | 3,781 (22.2%) | 723 (30.4%) | 2,891 (20.7%) |
| **Type of ventilation** | | | |
| Invasive ventilation only (IMV) | 7,235 (42.5%) | 1,772 (74.6%) | 5,105 (36.5%) |
| Non-invasive ventilation only (NIV) | 4,469 (26.3%) | 221 (9.3%) | 4,125 (29.5%) |
| Non-invasive ventilation failure (NIV-F) | 3,472 (20.4%) | 214 (9.0%) | 3,156 (22.5%) |
| Duration of ventilation between 1–6 hours | 1,152 (6.8%) | 85 (3.6%) | 1,026 (7.3%) |
| No ventilation procedure code | 695 (4.1%) | 84 (3.5%) | 586 (4.2%) |
| **In-hospital mortality** | 9,066 (53.3%) | 1,204 (50.7%) | 7,607 (54.3%) |

received invasive MV directly without having previously received NIV as a first escalation step. In contrast, only 37% received immediate invasive MV during the second pandemic wave. Consequently, more patients were escalated from oxygen therapy to NIV during the second

period (Table 1) with both patients successfully treated with NIV increasing from 9% to 30% and those with NIV-F increasing from 9% to 23%. However, the overall NIV-F rate decreased from 49% (214 of 435) to 43% (3156 of 7281).

The overall mortality rate of patients receiving any form of MV in the first and second wave of the pandemic increased from 51% and 54%. Overall mortality rates were lower for patients receiving NIV only (44%) compared to those with invasive MV only (54%), as illustrated in more detail in Fig 1A. However, mortality rates of patients with NIV-F were highest (66%). Of note, the mortality rate in patients with NIV-F increased steadily, from 32% in patients with NIV-F on the first day to 72% in those with NIV-F on day 5 or later (Fig 1B).

Overall, 7.941 patients had initially received NIV, with 3.472 having failed (NIV-F rate 44%) (Table 2). The highest proportion of NIV-F was found in the age group between 60–79 years. The NIV-F rate was lower in women (31%) than in men (69%). There was no clear

A

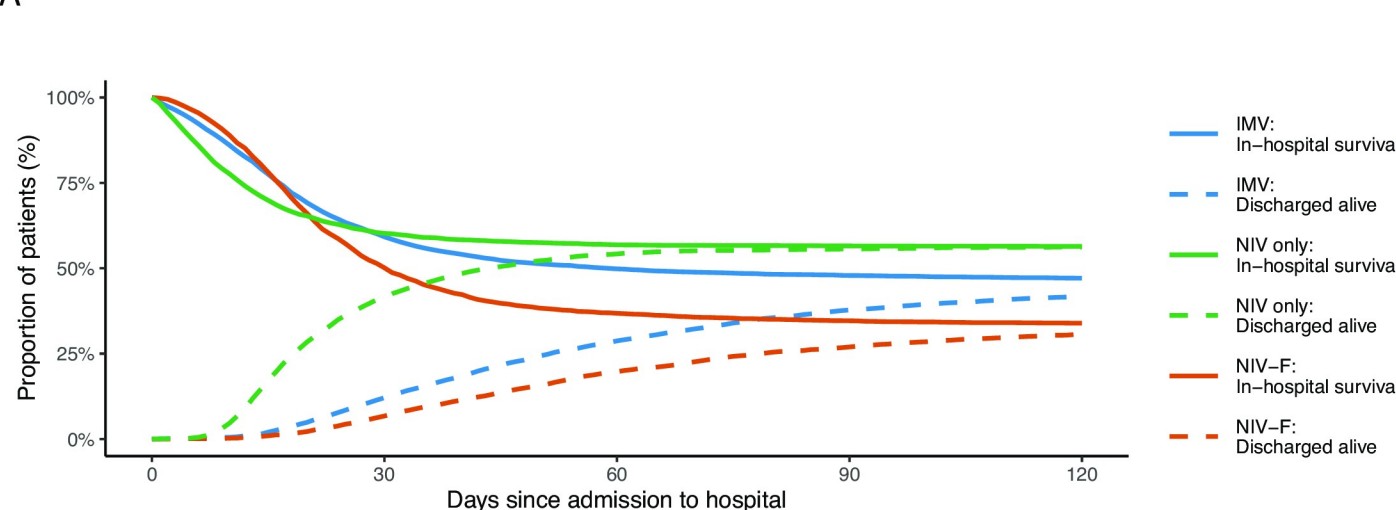

B

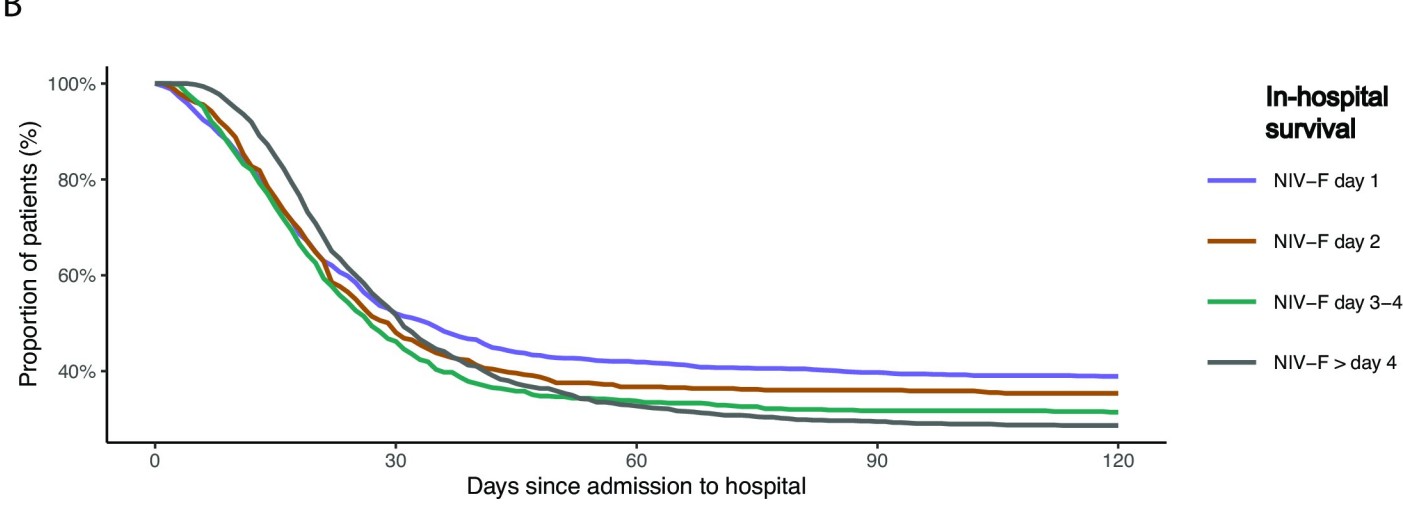

**Fig 1.** A. In-hospital mortality by type of ventilation. IMV = invasive mechanical ventilation (n = 3851), NIV = non-invasive-ventilation failure (n = 1614) and NIV-F = non-invasive-ventilation failure (n = 1247). B. In-hospital mortality of non-invasive-ventilation failure (NIV-F, n = 1247) by day of intubation.

**Table 2. Patient characteristics by type of ventilation.**

| | Invasive ventilation only (IMV) | Non-invasive ventilation only (NIV) | Non-invasive ventilation failure (NIV-F) | Non-invasive ventilation failure by day of failure | | | |
| --- | --- | --- | --- | --- | --- | --- | --- |
| | | | | First day | Second day | Third or fourth day | Fifth day or later |
| **Number of patients** | 7235 | 4469 | 3472 | 1213 | 591 | 684 | 984 |
| **Age (years)** | | | | | | | |
| Mean (SD) | 68.0 (12.9) | 71.2 (13.5) | 68.6 (11.9) | 67.9 (12.5) | 68.6 (12.3) | 68.9 (11.4) | 69.1 (11.0) |
| Median (IQR) | 70.0 (60.0, 78.0) | 74.0 (62.0, 82.0) | 70.0 (62.0, 78.0) | 69.0 (60.0, 78.0) | 70.0 (61.0, 78.0) | 71.0 (62.0, 78.0) | 70.0 (62.0, 78.0) |
| 18–49 years | 613 (8.5%) | 325 (7.3%) | 228 (6.6%) | 101 (8.3%) | 42 (7.1%) | 38 (5.6%) | 47 (4.8%) |
| 50–59 years | 1,095 (15.1%) | 541 (12.1%) | 470 (13.5%) | 176 (14.5%) | 78 (13.2%) | 88 (12.9%) | 128 (13.0%) |
| 60–69 years | 1,904 (26.3%) | 886 (19.8%) | 996 (28.7%) | 336 (27.7%) | 173 (29.3%) | 197 (28.8%) | 290 (29.5%) |
| 70–79 year | 2,135 (29.5%) | 1,131 (25.3%) | 1,102 (31.7%) | 358 (29.5%) | 178 (30.1%) | 230 (33.6%) | 336 (34.1%) |
| ≥ 80 years | 1,488 (20.6%) | 1,586 (35.5%) | 676 (19.5%) | 242 (20.0%) | 120 (20.3%) | 131 (19.2%) | 183 (18.6%) |
| **Male** | 4,687 (64.8%) | 2,725 (61.0%) | 2,405 (69.3%) | 816 (67.3%) | 418 (70.7%) | 475 (69.4%) | 696 (70.7%) |
| **Female** | 2,548 (35.2%) | 1,744 (39.0%) | 1,067 (30.7%) | 397 (32.7%) | 173 (29.3%) | 209 (30.6%) | 288 (29.3%) |
| **Elixhauser comorbidities** | | | | | | | |
| Hypertension | 4,913 (67.9%) | 3,129 (70.0%) | 2,482 (71.5%) | 839 (69.2%) | 428 (72.4%) | 476 (69.6%) | 739 (75.1%) |
| Diabetes | 3,182 (44.0%) | 1,889 (42.3%) | 1,556 (44.8%) | 543 (44.8%) | 258 (43.7%) | 302 (44.2%) | 453 (46.0%) |
| Cardiac arrhythmias | 3,304 (45.7%) | 1,619 (36.2%) | 1,641 (47.3%) | 557 (45.9%) | 290 (49.1%) | 327 (47.8%) | 467 (47.5%) |
| Renal failure | 2,006 (27.7%) | 1,430 (32.0%) | 931 (26.8%) | 323 (26.6%) | 172 (29.1%) | 185 (27.0%) | 251 (25.5%) |
| Congestive heart failure | 2,494 (34.5%) | 1,510 (33.8%) | 1,195 (34.4%) | 398 (32.8%) | 212 (35.9%) | 238 (34.8%) | 347 (35.3%) |
| Chronic pulmonary disease | 1,351 (18.7%) | 950 (21.3%) | 716 (20.6%) | 230 (19.0%) | 118 (20.0%) | 151 (22.1%) | 217 (22.1%) |
| Obesity | 1,220 (16.9%) | 610 (13.6%) | 624 (18.0%) | 232 (19.1%) | 111 (18.8%) | 126 (18.4%) | 155 (15.8%) |
| Charlson comorbidity index: 0 | 1,192 (16.5%) | 953 (21.3%) | 613 (17.7%) | 226 (18.6%) | 109 (18.4%) | 115 (16.8%) | 163 (16.6%) |
| Charlson comorbidity index: 1 | 1,479 (20.4%) | 959 (21.5%) | 722 (20.8%) | 252 (20.8%) | 104 (17.6%) | 151 (22.1%) | 215 (21.8%) |
| Charlson comorbidity index: 2 | 1,179 (16.3%) | 725 (16.2%) | 620 (17.9%) | 220 (18.1%) | 107 (18.1%) | 124 (18.1%) | 169 (17.2%) |
| Charlson comorbidity index: 3–4 | 1,854 (25.6%) | 1,050 (23.5%) | 890 (25.6%) | 319 (26.3%) | 157 (26.6%) | 185 (27.0%) | 229 (23.3%) |
| Charlson comorbidity index: ≥ 5 | 1,531 (21.2%) | 782 (17.5%) | 627 (18.1%) | 196 (16.2%) | 114 (19.3%) | 109 (15.9%) | 208 (21.1%) |
| **Patients transferred between hospitals** | 2,963 (41.0%) | 800 (17.9%) | 1,397 (40.2%) | 501 (41.3%) | 228 (38.6%) | 267 (39.0%) | 401 (40.8%) |
| **Length of hospital stay (days)** | | | | | | | |
| Mean (SD) | 41.5 (40.1) | 20.4 (16.8) | 37.6 (33.6) | 35.6 (32.3) | 35.6 (32.0) | 34.7 (29.9) | 43.2 (37.7) |
| Median (IQR) | 29.0 (16.0, 53.5) | 16.0 (10.0, 26.0) | 27.0 (16.0, 47.0) | 26.0 (15.0, 45.0) | 26.0 (16.0, 44.5) | 25.0 (15.0, 44.2) | 30.0 (19.0, 54.0) |
| **Ventilation (days)** | | | | | | | |
| Mean (SD) | 19.0 (19.2) | 4.1 (4.7) | 21.6 (20.2) | 20.1 (19.5) | 20.6 (19.5) | 21.5 (20.3) | 24.2 (21.3) |
| Median (IQR) | 13.3 (6.3, 25.1) | 2.7 (1.2, 5.3) | 15.8 (9.2, 27.5) | 14.3 (8.2, 25.7) | 15.5 (8.9, 25.3) | 15.6 (9.3, 26.4) | 18.0 (10.5, 31.0) |
| **Tracheostomy** | 2,705 (37.4%) | 0 (0.0%) | 1,312 (37.8%) | 437 (36.0%) | 216 (36.5%) | 256 (37.4%) | 403 (41.0%) |
| **ECMO** | 616 (8.5%) | 14 (0.3%) | 479 (13.8%) | 155 (12.8%) | 71 (12.0%) | 90 (13.2%) | 163 (16.6%) |
| **Dialysis** | 2,130 (29.4%) | 257 (5.8%) | 1,188 (34.2%) | 396 (32.6%) | 201 (34.0%) | 240 (35.1%) | 351 (35.7%) |
| **In-hospital mortality** | 3,874 (53.5%) | 1,949 (43.6%) | 2,306 (66.4%) | 746 (61.5%) | 382 (64.6%) | 470 (68.7%) | 708 (72.0%) |

ECMO = extracorporeal membrane oxygenation, NIV = non-invasive ventilation, NIV-F = non-invasive ventilation failure.

trend for the influence of comorbidities both on the decision to intubate the patient immediately and on the risk of NIV-F, i.e. the proportion of patients with a specific comorbidity was similar in both groups. One exception refers to cardiac arrhythmias, which were lowest in patients successfully treated with NIV (Table 2).

The duration of MV was clearly dependent on its modality (Table 2). The median duration of MV was 2.7 days in those receiving NIV only but reached 14 days in those who were intubated directly. Of note, patients who were switched from initial NIV to invasive MV following NIV failure spent the longest periods on MV (median 16 days). This trend was also true for the application of ECMO, which was reported in 14% of NIV-F patients, compared to 9% in patients who were intubated without having initially received NIV. Importantly, the proportion of patients with late NIV-F (after 5 days or more of NIV followed by intubation) substantially increased during the second wave, as displayed in Fig 2.

## Discussion

The current analysis of 17.023 patients represents the largest case series of COVID-19 patients requiring NIV or invasive MV and shows significant differences between the spring and autumn/winter periods 2020/21 with regard to the modality of MV. The main findings are as follows: Firstly, there was a significant increase in the utilization of NIV in Germany during the second period. Accordingly, the proportion of patients with acute respiratory failure who were directly intubated decreased from 75% to 37%. This was associated with a reduced overall duration of MV, and length of hospital stay. Secondly, the NIV-F rate was still high, even though there was a trend for a lower NIV-F rate during the second period (42%) compared to the first period (49%). Thirdly, the overall mortality rate in patients requiring MV remains high at 54%. Fourthly, NIV-F was associated with an increased ECMO utilization, increased overall duration of MV and increased mortality, and this was particularly true for late NIV-F occurring 5 days or later following NIV initiation.

Several clinical considerations can be derived from the current findings. Most importantly, the present analysis shows that NIV has been clearly established in the treatment of severe respiratory failure attributable to COVID-19 in a real-life setting without resource limitations,

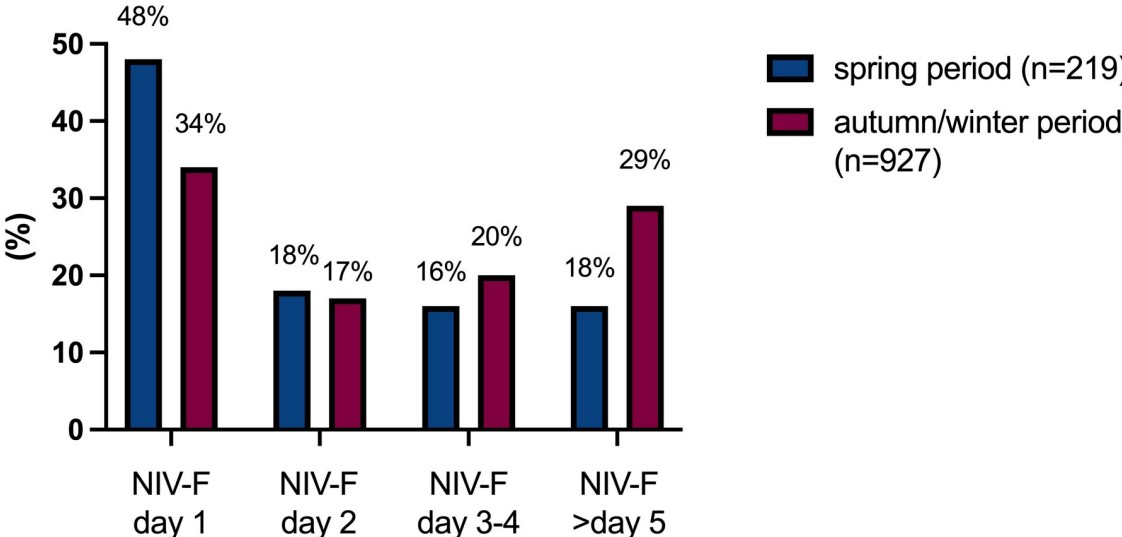

**Fig 2. Time distribution of NIV failure (NIV-F) by day, comparing spring and autumn period.**

since this is shown on a daily basis in the nationwide ICU registry, which counts all free and occupied ICU beds and COVID patients respectively (www.intensivregister.de). Thereby, in 2020 the overall duration of MV and hospital stay could be shortened. The decreasing NIV-F rate also suggests a learning curve that has occurred over the course of the last year, but may also be related to treatment successes outside MV, such as corticosteroids or prone positioning even with NIV at least in some centers [27, 28]. Remarkably, also the need of renal replacement therapy markedly dropped. Here, also steroids and/ or optimized treatment strategies, especially on mechanical ventilation may have attributed to this finding.

The present analysis, however, also demonstrates that clinicians should apply NIV cautiously as NIV-F continues to occur frequently, which is associated with increased mortality. Although reasons for intubation after NIV were not examined in this study, we did observe that a substantial proportion of patients initially treated with NIV progressed to receipt of IMV. The rather short median duration of NIV of 2.7 days in those patients successfully treated by NIV suggests that early improvements in respiratory function following NIV identify those patients who have been successfully treated and do not need intubation. In contrast, a longer duration of NIV, particularly exceeding 3–5 days, increases the likelihood of NIV-F, which is associated with an increased mortality. However, although administrative data in general contain no information e.g. about do not intubated or do not resuscitate order, the definition of NIV failure remained the same over the entire period of the study.

There are many other reports in the literature also showing the potential of NIV in the treatment of COVID-19-associated respiratory failure [29–31], and this might also have encouraged clinicians to more frequently and extensively apply NIV in this setting. In these reports, NIV was reasonably used outside the ICU, in part aimed at overcoming the shortage of ICU capacities [31]. Another rationale to use NIV as long as possible, also in the ICU setting, is aimed at avoiding intubation and intubation-related complications, most importantly lung injury related to invasive MV and infectious complications [9].

In this context, however, clinicians are likely to be less aware of a phenomenon related to maintained spontaneous breathing, which is labeled as patient self-inflicted lung injury (P-SILI) [32]. In short, initial lung injury related to COVID-19 is perpetually maintained and even aggravated as a consequence of a vicious circle that includes the sequence of capillary leakage, pulmonary edema, impaired gas exchange and respiratory mechanics, subsequent increase in respiratory drive followed by increased pleural pressure swings, which eventually lead to capillary leakage again if lowering of pleural pressure exceeds the intravascular pressure decrease [33, 34]. Even though the current data does not provide evidence for P-SILI in those patients having failed NIV, this phenomenon might, nevertheless, explain why outcome is severely reduced in patients spending longer durations on NIV, which eventually fails. Of note, the current data demonstrate, that women show independently of the first or second wave a less severe course of the disease.

## Limitations

There are some important limitations of the present data, which need to be addressed in the context of data interpretation. First, all data refer to the coding of diseases (ICD) and procedures (OPS) in the context of remuneration. Thus, patients were not studied directly. Therefore, several important data are missing, i.e. disease severity related to the $PaO_2/FiO_2$ ratio, intubation criteria, ventilator settings/equipment and oxygen flow rates including information on the response to treatment, "do-not-intubate" orders or details on medication.

Second, the analysis includes only data from one health care insurance company. However, this is the largest insurance and accounts for about 1/3 of the total population, providing a

large representative sample for the German population. Third, NIV as defined for the reimbursement system in Germany excludes high-flow oxygen treatment (HFOT) and continuous positive airway pressure (CPAP) and can only be coded if the level of pressure support exceeds 5 cm $H_2O$. Therefore, this analysis focused on non-invasive and invasive ventilation, which are both very accurately documented since reimbursement in ICU medicine in Germany mainly depends on MV. In addition, the German guidelines have recommended using HFOT as first escalation step when oxygen treatment is insufficient, while CPAP and NIV form the following escalation steps. Thus, NIV in the present analysis represents a rather selected group of patients, and this group may not be compared to studies from other countries without considering this aspect. Furthermore, patients with less than 6 documented hours of ventilation and patients with more than 6 documented hours of ventilation but without a corresponding procedure code for NIV or invasive ventilation were not assigned to the three subgroups. However, the patient number is low in these groups. Finally, since many factors are likely undetectable in the administrative dataset, unmeasured confounding and confounding by indication and over-or under-reporting or misclassification of cases remain additional major limitations of the study. These challenges are best addressed with a multicenter and multinational clinical trial that randomizes patients to NIV vs. IMV, with clear clinical criteria to standardize crossover to IMV.

## Conclusions

The utilization of NIV rapidly increased during the autumn/winter period compared to the spring period 2020 of the COVID-19 pandemic in Germany. This was associated with an overall reduced duration of MV, and length of hospital stay. However, the current data do not explain in detail the reasons of mortality, since also other treatment modifications may have contributed to the outcome. Despite this, overall mortality of patients receiving MV due to COVID-19-associated respiratory failure remained high at 53%. Patients successfully treated with NIV had lower mortality rates than those who were intubated directly, but those failing NIV had a higher mortality rate, respectively, and this became even more predominant in late NIV failure. Thus, the current study shows the increasing role of NIV in the context of ICU medicine related to COVID-19 during the second wave and, may also emphasize on its risks. Prompt identification of patients failing an NIV approach is mandatory to avoid harmful delays and very poor outcomes. Given these findings, there is a need for prospective randomized controlled trials that focus on the most reasonable indications for initiation of NIV as well as timely subsequent intubation in case of NIV failure in COVID-19 patients.

## Author Contributions

**Conceptualization:** Christian Karagiannidis, Corinna Hentschker, Michael Westhoff, Steffen Weber-Carstens, Uwe Janssens, Stefan Kluge, Michael Pfeifer, Claudia Spies, Tobias Welte, Rolf Rossaint, Carina Mostert, Wolfram Windisch.

**Data curation:** Christian Karagiannidis, Corinna Hentschker, Carina Mostert.

**Formal analysis:** Christian Karagiannidis, Corinna Hentschker, Michael Westhoff, Steffen Weber-Carstens, Carina Mostert, Wolfram Windisch.

**Methodology:** Christian Karagiannidis, Wolfram Windisch.

**Project administration:** Christian Karagiannidis.

**Supervision:** Christian Karagiannidis.

**Validation:** Christian Karagiannidis.

**Visualization:** Christian Karagiannidis.

**Writing – original draft:** Christian Karagiannidis, Corinna Hentschker, Michael Westhoff, Steffen Weber-Carstens, Uwe Janssens, Stefan Kluge, Michael Pfeifer, Rolf Rossaint, Carina Mostert, Wolfram Windisch.

**Writing – review & editing:** Michael Pfeifer, Claudia Spies, Tobias Welte, Rolf Rossaint, Wolfram Windisch.

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
