## [Decision Letter · Decision Letter 0]

28 Aug 2021

PONE-D-21-23922

Observational study of changes in utilization and outcomes in non-invasive ventilation in COVID-19

PLOS ONE

Dear Dr. Karagiannidis,

Thank you for submitting your manuscript to PLOS ONE. After careful consideration, we feel that it has merit but does not fully meet PLOS ONE’s publication criteria as it currently stands. Therefore, we invite you to submit a revised version of the manuscript that addresses the points raised during the review process.

ACADEMIC EDITOR: I support the comments from the Reviewers, that are two experts in the field. Specifically, more details on the methods and on statistical plan and approach should be provided to re-evaluate the paper.

We look forward to receiving your revised manuscript.

Kind regards,

Andrea Cortegiani, M.D.

Academic Editor

PLOS ONE

Journal Requirements:

"Dr. Karagiannidis reports personal fees from Maquet, personal fees from Xenios, personal fees from Bayer, non-financial support from Speaker of the German register of ICUs, grants from German Ministry of Research and Education, during the conduct of the study. Dr. Hentschker has nothing to disclose. Dr. Westhoff has nothing to disclose. Dr. Weber-Carstens has nothing to disclose. Dr. Janssens has nothing to disclose. Dr. Kluge reports non-financial support from Ambu, ETView Ltd, Fisher & Paykel and Xenios., grants from Daiichi Sankyo, Pfizer, personal fees from Astra, C.R. Bard, Baxter, Biotest, Cytosorbents, Fresenius, Gilead, MSD, Pfizer, Philips, ZOLL, personal fees and other from Bayer, Fresenius, Gilead, MSD und Pfizer, outside the submitted work. MP reports no conflicts of interests in regard to the manuscript, lecture fees from Boehringer, Novartis, Astra_Zeneca, Roche and fees for advisory board meetings from Boehringer, Novartis, Roche Current president of the German Society of Pneumology. Dr. Spies reports grants from Public Grants, grants from IIT grants from companies, other from Meeting support from companies (e.g. for the Leopoldina 2020 meeting), outside the submitted work; In addition, Dr. Spies has a patent EEG monitoring licensed, and a patent Ceilings licensed. Dr. Welte reports grants from German Minstry of Research and Education, during the conduct of the study."

Reviewers' comments:

Reviewer's Responses to Questions

**Comments to the Author**

1. Is the manuscript technically sound, and do the data support the conclusions?

Reviewer #1: No

Reviewer #2: Yes

2. Has the statistical analysis been performed appropriately and rigorously? 

Reviewer #1: No

Reviewer #2: Yes

3. Have the authors made all data underlying the findings in their manuscript fully available?

Reviewer #1: Yes

Reviewer #2: Yes

4. Is the manuscript presented in an intelligible fashion and written in standard English?

Reviewer #1: Yes

Reviewer #2: Yes

5. Review Comments to the Author

Reviewer #1: I would like to thank the Editor for the chance to review this work by Karagiannidis and co-workers.

The authors present a considerable amount of interesting data regarding the changes in use of mechanical ventilation in COVID-19 patients in two pandemic waves during 2020. They report the use of NIV rapidly increased during the second wave without change in overall mortality.

General comment

The question is appealing as the use of NIV in hypoxic patients (including COVID-19) still remains a hot topic. My personal opinion is that, although data are interesting given the considerable number of patients enrolled, several methodological issues negatively affect the quality of the work. In particular the lack of a pre-specified statistical approach to this huge amount of data and the absence of a solid method to account for confounders in the analysis, do not allow to draw conclusions.

Specific comments

Title

The title refers to the change in use and outcome of NIV. However, a consistent amount of data and analyses focus on the use of invasive mechanical ventilation in patients that did not receive NIV trial. This should be reported in the title. (i.e. Observational study of changes in utilization and outcomes in mechanical ventilation in COVID-19) otherwise analysis should be limited only to NIV use over time.

Introduction

Line 129. Is the analysis only limited to the intensive care setting? During the pandemic NIV has been extensively used even outside the ICU. It would be important to know if these data refer to the use of NIV in general or only limited to the ICU setting (where technological monitoring, trained nursing and specific critical care skills are far more granted).

Methods

The methods section should be expanded to illustrate the pre-specified analyses performed once data had been categorized (invasive MV, NIV, NIV-F). What statistical approach was applied? Which inferential methodology? This information is essential to understand the nature of this investigation and the reliability of findings.

Line 142. I am not sure this would be a reliable method to assess NIV duration as it may be discontinued over time and then resumed for clinical worsening. Consider adding this point in the limitations section.

Discussion

As reported in the limitations section, one of the major limitations is the lack of a pre-specified definition of NIV failure. Was this definition the same in the two pandemic periods?More details should be provided on this topic.

How did the authors account for NIV patients with ceiling of escalation to intensive care? This subgroup of patients might have affected the results of NIV mortality.

Conclusion

I am not sure conclusions regarding mortality are sufficiently supported by the presented results as many confounding factors were not included in the analysis.

Reviewer #2: GENERAL COMMENTS

Thank you for allowing me to review this interesting manuscript. This is a retrospective study of administrative (healthcare insurance) database analysis showing NIV utilization in patients with SARS-CoV-2 infection in Germany. The authors concluded that successful NIV lowered mortality rates, but NIV failure was associated with greater mortality risk. Therefore, prompt identification of those failing an NIV approach is mandatory to avoid harmful delays and very poor outcomes. The manuscript deals with a clinically relevant topic, given that the role of NIV is still debated in COVID-19 ARDS.

SPECIFIC COMMENTS

The quality of written English is acceptable.

Major comment

Abstract:

1) Aim: Please specify that you also want to explore differences of patients treated with NIV between first and second wave

Introduction:

The background is concise and informative. However, I have some suggestions:

1) Line 89 this statement needs a reference.

2) Line 94 – 95 “Non-invasive ventilation (NIV) is suggested to reduce the complications of invasive MV(9)”. I suggest to state that the use of noninvasive respiratory support in acute respiratory failure due to viral infection is still debated quoting as reference a systematic review to help the readers better understand the context (e.g. doi: 10.23736/S0375-9393.20.14785-0)

3) Line 95-98: I suggest to add that Guidelines from different regions on the use of NIV in COVID-19 have been inconsistent and heterogeneous quoting this reference doi: 10.1016/j.ijid.2021.03.078

4) Lines 98-101 you state that “personal communication suggests that face masks are by far the most widely used interfaces”. What do you mean by "personal communication"? Please clarify. This statement needs a reference, as reported, it looks like an opinion. You might state that face masks are the most commonly used interface in acute respiratory failure in clinical practice in Europe, quoting this survey doi:10.1183/09031936.00123509

5) Lines 109-113 Aim: I would add that you also want to explore practice changes between spring and autumn period.

.

Methods:

The methods used are appropriate for the retrospective design of the study based on data registry. Study question is clearly stated and clinically relevant. However, I have some remarks:

1) Please specify that this is a retrospective analysis of claim data from registry.

2) Who did perform data extraction from the insurance database? research staff of the insurance company or study investigator? Please specify.

3) Please clarify in the methods section how you did account for comorbidities. I see in Table 1 that you used "Elixhauser comorbidities", please add this information in the methods section and provide a reference for "Elixhauser comorbidities". Moreover, I would suggest also adding the Charlson comorbidity index, which is much more used in clinical practice and in clinical research and it may be more helpful for the reader to interpret the data from the study.

4) Please specify the ICD procedure code used to identify the use of NIV and IMV.

5) In your search, was COVID-19 the primary admission diagnosis code for hospitalization from ICD or one of the secondary admission diagnoses? Please specify this aspect. Selected patients with a different (non-COVID) primary admitting diagnosis might also require NIV or IMV and develop COVID during hospitalization (especially in the first wave of pandemic).

RESULTS

1) A study flowchart to illustrate included/excluded patients would be helpful to the reader to explain the flow of patients in the study

2) Table 1: I would add the overall NIV-F rate number of patients who received NIV prior to ICU admission (Noninvasive ventilation on ICU admission)

DISCUSSION

The discussion is balanced. References are relevant and updated. Limitations of the study are correctly addressed by the authors and discussed, but I would recommend emphasizing some aspects:

1) Please emphasize that, as with all data routinely collected for other purposes (health insurances), the accuracy and completeness of the information may be compromised due to over-or under-reporting or misclassification of cases.

2) Ventilation days, tracheostomy, and dialysis differed between the 2 periods (nearly 10% less during the second wave). Were patients of the second wave less severe? Please discuss this point.

3) How would you explain the lack of improvement in intensive care mortality or ventilated patients' prognosis despite the decreasing NIV-F rate in the second wave?

4) How would you explain the change in clinical practice on NIV use between the two periods? Please discuss this aspect.

5) Was awake prone positioning during NIV part of the usual clinical practice in Germany? If yes, please add this intervention in lines 213-215 "The decreasing NIV-F rate also suggests a learning curve that has occurred over the course of the last year, but may also be related to treatment successes outside MV, such as corticosteroids (24, 25)".

6) Line 252-253 In the limitation settings you state that is not possible to identify the hospital settings (ICU, intermediate care, COVID-19 wards) but in the introduction lines 109-110 you state that the aim of the study is "to determine detailed characteristics and outcomes of 7,490 hospitalized COVID-19 patients with MV on the ICU", please clarify.

7) I would emphasize more the message that prompt identification of patients failing an NIV approach is mandatory to avoid harmful delays and very poor outcomes.

Minor comments:

1) Table 1 “elixhauser” please report the full definition in the table legend

2) Line 210-211 please check punctuation

6. PLOS authors have the option to publish the peer review history of their article (what does this mean?). If published, this will include your full peer review and any attached files.

Reviewer #1: No

Reviewer #2: No

---

## [Author Response · Author response to Decision Letter 0]

16 Oct 2021

Response to Reviewer 1: We appreciate the very thoughtful critiques by the reviewers, and we have revised our manuscript accordingly. Furthermore, we updated the data extensively until February 2021, which remarkably increased the patient number of the study from 7000 to 17000. 

Reviewer #1: I would like to thank the Editor for the chance to review this work by Karagiannidis and co-workers.

The authors present a considerable amount of interesting data regarding the changes in use of mechanical ventilation in COVID-19 patients in two pandemic waves during 2020. They report the use of NIV rapidly increased during the second wave without change in overall mortality.

General comment

The question is appealing as the use of NIV in hypoxic patients (including COVID-19) still remains a hot topic. My personal opinion is that, although data are interesting given the considerable number of patients enrolled, several methodological issues negatively affect the quality of the work. In particular the lack of a pre-specified statistical approach to this huge amount of data and the absence of a solid method to account for confounders in the analysis, do not allow to draw conclusions.

Thank you very much for your effort. We revised our manuscript accordingly. Furthermore, we updated the patient data and included now all patients from the second wave until February 2021. This increased the patient number substantially.

Specific comments

Title

The title refers to the change in use and outcome of NIV. However, a consistent amount of data and analyses focus on the use of invasive mechanical ventilation in patients that did not receive NIV trial. This should be reported in the title. (i.e. Observational study of changes in utilization and outcomes in mechanical ventilation in COVID-19) otherwise analysis should be limited only to NIV use over time.

Thank you very much for this very important point. We changed the title accordingly.

Introduction

Line 129. Is the analysis only limited to the intensive care setting? During the pandemic NIV has been extensively used even outside the ICU. It would be important to know if these data refer to the use of NIV in general or only limited to the ICU setting (where technological monitoring, trained nursing and specific critical care skills are far more granted).

Our Data are limited to the ICU setting. In Germany only very few patients are treated outside the ICU with NIV. However, some of these patients are treated on so-called “low-care ICU beds”. We added this point to the methods: “The primary analysis includes all patients with mechanical ventilation, either non-invasive or invasive on the ICU without any missing”

Methods

The methods section should be expanded to illustrate the pre-specified analyses performed once data had been categorized (invasive MV, NIV, NIV-F). What statistical approach was applied? Which inferential methodology? This information is essential to understand the nature of this investigation and the reliability of findings.

We added some more specific comments to the Methods part now: “Therefore, cases who were transferred during their hospital stay (discharge date of one hospital corresponds to the admission date of another hospital) were merged. Thus, the current analysis was performed at the patient level. The following OPS codes from the German DRG systematic coding were analyzed: 8-701, 8-704, 5-311, 5-313 and 8-706. For continuous variables, we report means with SDs and medians with IQRs. For categorical variables, we report absolute numbers and percentages.”

Line 142. I am not sure this would be a reliable method to assess NIV duration as it may be discontinued over time and then resumed for clinical worsening. Consider adding this point in the limitations section.

We added this point accordingly in the limitation section: “Furthermore, patients with less than 6 documented hours of ventilation and patients with more than 6 documented hours of ventilation but without a corresponding procedure code for NIV or invasive ventilation were not assigned to the three subgroups. However, the patient number is low in these groups”

Discussion

As reported in the limitations section, one of the major limitations is the lack of a pre-specified definition of NIV failure. Was this definition the same in the two pandemic periods? More details should be provided on this topic.

We clarified this topic in the discussion section: “However, although administrative data in general contain no information e.g. about do not intubated or do not resuscitate order, the definition of NIV failure remained the same over the entire period of the study.”

How did the authors account for NIV patients with ceiling of escalation to intensive care? This subgroup of patients might have affected the results of NIV mortality.

In Germany only few patients with NIV are treated outside the ICU. The reason for it is, that reimbursement is dependent on being on the ICU and then counting the numbers of hours on MV. We would be happy if we may not address this point in the manuscript.

Conclusion

I am not sure conclusions regarding mortality are sufficiently supported by the presented results as many confounding factors were not included in the analysis.

We agree and integrate the following sentence to the conclusion: “However, the current data do not explain in great detail the reasons of mortality, since also other treatment modifications may have contributed to the outcome.”

Response to Reviewer 1: We appreciate the very thoughtful critiques by the reviewers, and we have revised our manuscript accordingly. Furthermore, we updated the data extensively until February 2021, which remarkably increased the patient number of the study from 7000 to 17000. 

Reviewer #2: GENERAL COMMENTS

Thank you for allowing me to review this interesting manuscript. This is a retrospective study of administrative (healthcare insurance) database analysis showing NIV utilization in patients with SARS-CoV-2 infection in Germany. The authors concluded that successful NIV lowered mortality rates, but NIV failure was associated with greater mortality risk. Therefore, prompt identification of those failing an NIV approach is mandatory to avoid harmful delays and very poor outcomes. The manuscript deals with a clinically relevant topic, given that the role of NIV is still debated in COVID-19 ARDS.

SPECIFIC COMMENTS

The quality of written English is acceptable.

Major comment

Abstract:

Aim: Please specify that you also want to explore differences of patients treated with NIV between first and second wave

We are thankful for this comment and integrated this comment now: “Furthermore, differences of patients treated with NIV between the first and second wave are explored.”

Introduction:

The background is concise and informative. However, I have some suggestions:

1) Line 89 this statement needs a reference.

We added this reference now: “Within one year, the SARS-CoV-2 pandemic has affected more than 235 million people worldwide (https://coronavirus.jhu.edu/map.html).”

2) Line 94 – 95 “Non-invasive ventilation (NIV) is suggested to reduce the complications of invasive MV(9)”. I suggest to state that the use of noninvasive respiratory support in acute respiratory failure due to viral infection is still debated quoting as reference a systematic review to help the readers better understand the context (e.g. doi: 10.23736/S0375-9393.20.14785-0)

We revised it accordingly: “The use of noninvasive respiratory support in acute respiratory failure due to viral infection is still debated (9){Crimi, 2020 #2307}”

3) Line 95-98: I suggest to add that Guidelines from different regions on the use of NIV in COVID-19 have been inconsistent and heterogeneous quoting this reference doi: 10.1016/j.ijid.2021.03.078

We added this very valuable point now: “despite international guidelines being still inhomogeneous in recommendations (18)”

4) Lines 98-101 you state that “personal communication suggests that face masks are by far the most widely used interfaces”. What do you mean by "personal communication"? Please clarify. This statement needs a reference, as reported, it looks like an opinion. You might state that face masks are the most commonly used interface in acute respiratory failure in clinical practice in Europe, quoting this survey doi:10.1183/09031936.00123509

Thak you very much, we changed this accordingly: “In Germany, detailed epidemiological data about what types of interfaces are used are not available, but face masks are the most commonly used interface in acute respiratory failure in clinical practice in Europe (19), while helmets are only rarely used by very few experienced centers”

5) Lines 109-113 Aim: I would add that you also want to explore practice changes between spring and autumn period.

We changed this accordingly in the aim.

.

Methods:

The methods used are appropriate for the retrospective design of the study based on data registry. Study question is clearly stated and clinically relevant. However, I have some remarks:

1) Please specify that this is a retrospective analysis of claim data from registry.

We added this now to this section: “In general this is a retrospective analysis of claims data from this registry.” 

2) Who did perform data extraction from the insurance database? research staff of the insurance company or study investigator? Please specify.

This was clearly specified now: “Data extraction was done by the scientific institute of the health care insurance, whereas analysis was done by the author group”

3) Please clarify in the methods section how you did account for comorbidities. I see in Table 1 that you used "Elixhauser comorbidities", please add this information in the methods section and provide a reference for "Elixhauser comorbidities". Moreover, I would suggest also adding the Charlson comorbidity index, which is much more used in clinical practice and in clinical research and it may be more helpful for the reader to interpret the data from the study.

We now added the Charlson comorbidity index to the table 1 and 2.

4) Please specify the ICD procedure code used to identify the use of NIV and IMV.

This was added to the Method section now:” Therefore, cases who were transferred during their hospital stay (discharge date of one hospital corresponds to the admission date of another hospital) were merged. Thus, the current analysis was performed at the patient level. The following OPS codes from the German DRG systematic coding were analyzed: 8-701, 8-704, 5-311, 5-313 and 8-706. For continuous variables, we report means with SDs and medians with IQRs. For categorical variables, we report absolute numbers and percentages.”

5) In your search, was COVID-19 the primary admission diagnosis code for hospitalization from ICD or one of the secondary admission diagnoses? Please specify this aspect. Selected patients with a different (non-COVID) primary admitting diagnosis might also require NIV or IMV and develop COVID during hospitalization (especially in the first wave of pandemic).

Most patients are admitted because of COVID-19. We did several analyses on this point by grouping the patients with and without primary respiratory insufficiency as far as you can do that from these data. All robustness checks revealed the same outcome. Therefore it seems that this point occurs, but does not really make a difference.2

RESULTS

1) A study flowchart to illustrate included/excluded patients would be helpful to the reader to explain the flow of patients in the study

We generally agree with this remark, but our search strategy was patients with confirmed COVID diagnosis and MV. Therefore from our perspective a flow chart would not really help the reader.

2) Table 1: I would add the overall NIV-F rate number of patients who received NIV prior to ICU admission (Noninvasive ventilation on ICU admission) 

Unfortunately we don´t have the data, but we expect only very, very few patients in this group, since NIV is done on high-care or low-care ICUs in Germany, but not outside. This is very different to other countries in the world.

DISCUSSION

The discussion is balanced. References are relevant and updated. Limitations of the study are correctly addressed by the authors and discussed, but I would recommend emphasizing some aspects:

1) Please emphasize that, as with all data routinely collected for other purposes (health insurances), the accuracy and completeness of the information may be compromised due to over-or under-reporting or misclassification of cases.

We added this point to the limitation section: “Finally, since many factors are likely undetectable in the administrative dataset, unmeasured confounding and confounding by indication and over-or under-reporting or misclassification of cases remain additional major limitations of the study.”

2) Ventilation days, tracheostomy, and dialysis differed between the 2 periods (nearly 10% less during the second wave). Were patients of the second wave less severe? Please discuss this point.

This is an important but unknown point. In the international discussion we had on this are two opinions: better treatment and/or steroids. Many countries observed this.We added this now to the discussion: “Remarkably, also the need of renal replacement therapy markedly dropped. Here, also steroids and/ or optimized treatment strategies, especially on mechanical ventilation may have attributed to this finding.”

3) How would you explain the lack of improvement in intensive care mortality or ventilated patients' prognosis despite the decreasing NIV-F rate in the second wave?

This is highly speculative. Overuse of NIV might be one problem, more bacterial superinfection might be another reason. Since we cannot prove it, we would not like to stress this point in the discussion.

4) How would you explain the change in clinical practice on NIV use between the two periods? Please discuss this aspect.

Indeed, the huge increase in NIV usage at the second wave is remarkable. Even though further explanation is needed, there was an increasingly insistent discussion on the rationale of NIV to be used in COVID-19 patients in the ICU setting that has started to emerge with the dwindling of the first pandemic wave.

5) Was awake prone positioning during NIV part of the usual clinical practice in Germany? If yes, please add this intervention in lines 213-215 "The decreasing NIV-F rate also suggests a learning curve that has occurred over the course of the last year, but may also be related to treatment successes outside MV, such as corticosteroids (24, 25)".

We recommended to use prone positioning in NIV in Octobre 2021. However, some centres did that even before. We added this now:” …such as corticosteroids or prone positioning even with NIV at least in some centers”

6) Line 252-253 In the limitation settings you state that is not possible to identify the hospital settings (ICU, intermediate care, COVID-19 wards) but in the introduction lines 109-110 you state that the aim of the study is "to determine detailed characteristics and outcomes of 7,490 hospitalized COVID-19 patients with MV on the ICU", please clarify.

Thanks for this comment. We removed this part from the limitation section, since this is misleading. We can identify the patients by a code for the ICU in Germany. This is what we did in the analysis. Sorry for the confusion.

7) I would emphasize more the message that prompt identification of patients failing an NIV approach is mandatory to avoid harmful delays and very poor outcomes.

We fully agree and added this point to the conclusions:” Prompt identification of patients failing an NIV approach is mandatory to avoid harmful delays and very poor outcomes.”

Minor comments:

1) Table 1 “elixhauser” please report the full definition in the table legend 

We integrated this into the table and legend now. 

2) Line 210-211 please check punctuation

Thanks for this very careful correction. This was done.

---

## [Decision Letter · Decision Letter 1]

21 Dec 2021

Observational study of changes in utilization and outcomes in mechanical ventilation in COVID-19

PONE-D-21-23922R1

Dear Dr. Karagiannidis,

We’re pleased to inform you that your manuscript has been judged scientifically suitable for publication and will be formally accepted for publication once it meets all outstanding technical requirements.

Kind regards,

Andrea Cortegiani, M.D.

Academic Editor

PLOS ONE

Additional Editor Comments (optional):

Reviewers' comments:

Reviewer's Responses to Questions

**Comments to the Author**

1. If the authors have adequately addressed your comments raised in a previous round of review and you feel that this manuscript is now acceptable for publication, you may indicate that here to bypass the “Comments to the Author” section, enter your conflict of interest statement in the “Confidential to Editor” section, and submit your "Accept" recommendation.

Reviewer #1: All comments have been addressed

Reviewer #2: All comments have been addressed

2. Is the manuscript technically sound, and do the data support the conclusions?

Reviewer #1: Yes

Reviewer #2: Yes

3. Has the statistical analysis been performed appropriately and rigorously? 

Reviewer #1: Yes

Reviewer #2: Yes

4. Have the authors made all data underlying the findings in their manuscript fully available?

Reviewer #1: Yes

Reviewer #2: Yes

5. Is the manuscript presented in an intelligible fashion and written in standard English?

Reviewer #1: Yes

Reviewer #2: Yes

6. Review Comments to the Author

Reviewer #1: I would like to thank the Authors for their work and the responses give to my concerns.

I do not have further comments to make.

Reviewer #2: The authors addressed all the points raised by the reviewers and should be congratulated for their effort. The manuscript has improved significantly. I have no additional remarks.

7. PLOS authors have the option to publish the peer review history of their article (what does this mean?). If published, this will include your full peer review and any attached files.

Reviewer #1: **Yes: **Roberto Tonelli

Reviewer #2: No

---

## [Editor Report · Acceptance letter]

5 Jan 2022

PONE-D-21-23922R1 

Observational study of changes in utilization and outcomes in mechanical ventilation in COVID-19 

Dear Dr. Karagiannidis:

I'm pleased to inform you that your manuscript has been deemed suitable for publication in PLOS ONE. Congratulations! Your manuscript is now with our production department. 

Kind regards, 

on behalf of

Dr. Andrea Cortegiani 

Academic Editor

PLOS ONE